# Quality Evaluation of *Ophiopogon japonicus* from Two Authentic Geographical Origins in China Based on Physicochemical and Pharmacological Properties of Their Polysaccharides

**DOI:** 10.3390/biom12101491

**Published:** 2022-10-16

**Authors:** Zherui Chen, Baojie Zhu, Xin Peng, Shaoping Li, Jing Zhao

**Affiliations:** 1State Key Laboratory of Quality Research in Chinese Medicine, Institute of Chinese Medical Sciences, University of Macao, Macao SAR 999078, China; 2Joint Laboratory of Chinese Herbal Glycoengineering and Testing Technology, University of Macao, Macao SAR 999078, China; 3Macao Centre for Testing of Chinese Medicine, University of Macao, Macao SAR 999078, China; 4Ningbo Municipal Hospital of Traditional Chinese Medicine, Affiliated Hospital of Zhejiang Chinese Medical University, Ningbo 315000, China

**Keywords:** *MaiDong* polysaccharides, quality control, physicochemical properties, pharmacological properties, multivariate statistical analysis

## Abstract

*Ophiopogon japonicus* is widely used as a tonic herb in China. According to the origins, MaiDong of Chinese materia medica can be classified as *Zhe MaiDong* (*Ophiopogon japonicus* in Zhejiang), *Chuan MaiDong* (*Ophiopogon japonicus* in Sichuan), *Duanting Shan MaiDong* (*Liriope muscari*), and *Hubei MaiDong* (*Liriope spicata*). In terms of quality control, polysaccharides-based evaluations have not yet been conducted. In this study, microwave-assisted extraction (MAE) was used for the preparation of polysaccharides from 29 batches of *MaiDong*. HPSEC-MALLS-RID and HPAEC-PAD were employed to investigate their molecular parameters and compositional monosaccharides, respectively. The ability to scavenge ABTS radicals and immune promotion abilities, in terms of nitric oxide releasing and phagocytosis on RAW 264.7 macrophages, were also compared. The results showed that polysaccharides in different *MaiDong* varied in molecular parameters. All polysaccharides mainly contained fructose and glucose with small amounts of arabinose, mannose, galactose, and xylose. For polysaccharides of *Zhe MaiDong* and *Chuan MaiDong*, the molar ratio of Fru to Glc was roughly 15:1 and 14:1, respectively. *Zhe MaiDong* exhibited better antioxidant and immune promotion activity, and so did that of fibrous roots. The pharmacological activity, however, did not account for the variation in growth years. Finally, indicators for quality control based on multivariate statistical analysis included: yield, antioxidant activity, the content of fructose, and RI signal. It was concluded that *MaiDong*’s fibrous roots had similar components to the root, and their quality was not significantly affected by growth age. This may provide some guidance for the cultivation and use of *MaiDong*.

## 1. Introduction

The demand for traditional Chinese medicines (TCMs) continues to grow alongside their modernization and globalization. The contradiction between the supply and demand of TCMs could be effectively alleviated by artificial cultivation [1,2]. Studies have shown that TCMs from diverse origins often differ in chemical properties and pharmacological effects [3,4,5]. Authentic (*Daodi* in Chinese) medicinal materials, TCMs with regional characteristics grown in a specific ecological environment, are thought to be of higher quality, according to traditional Chinese medicine theory [6,7].

*Ophiopogon japonicus* (called *MaiDong* in Chinese) is a tonic of the same origin as medicine and food and has a long history of use in China. It is widely used in clinical practice because it has the effect of nourishing the Yin and moistening the lung. Modern pharmacological research has demonstrated the effects of *MaiDong*, which include anti-oxidation, anti-tumor, anti-inflammatory, and cardiovascular protective properties [4,8,9,10,11,12]. Conventionally, Zhejiang (*Zhe MaiDong*) and Sichuan (*Chuan MaiDong*) provinces are two major authentic geographical origins of *O*. *japonicus* that originates in China. *Chuan MaiDong* can typically be harvested after growing for one year for medicinal purposes; however, *Zhe MaiDong* must grow for three years before harvest, resulting in greater cultivation expense. Therefore, their quality differences are the major concern in clinics [13]. To assess the quality of *Zhe MaiDong* and *Chuan MaiDong*, saponins, isoflavones, and other small molecules have been compared [11]. Furthermore, to compare *MaiDong* from various sources, the total homoisoflavonoid concentration [14], antioxidant activity [15], and anti-inflammatory activities of saponins [16] were evaluated. Additionally, the results indicated that those of *Zhe MaiDong* are usually higher than those of *Chuan MaiDong*.

Polysaccharides, with anti-glycemic, immunomodulatory, and gastrointestinal protective effects [4,12,17,18,19,20], are abundant in *MaiDong*. However, few studies have focused on the difference between *Zhe MaiDong* and *Chuan MaiDong* because of their complexity. In this study, physicochemical and pharmacological properties of polysaccharides from 29 batches of *Zhe MaiDong* and *Chuan MaiDong*, as well as *Duanting Shan MaiDong* (*Liriope muscari*) and *Hubei MaiDong* (*Liriope spicata*), were investigated and compared, which could be helpful to improve their quality control.

## 2. Materials and Methods

### 2.1. Materials and Chemicals

Raw materials of *O*. *japonicus*, *L. muscari* (named as *Shan MaiDong*), and *L*. *spicata* (named *Hubei MaiDong*) are listed in Table 1. Their species were identified by Professor Xin Peng, one of the authors of this paper. The samples were stored in the Institute of Chinese Medical Sciences, University of Macau.

2,2′-azino-bis (3-ethylbenzothiazoline-6-sulfonic acid) (ABTS) was purchased from International Laboratory (San Bruno, CA, USA). Potassium persulfate was purchased from Fluka (Selzer, Germany). Dulbecco’s modified eagle medium (DMEM), fetal bovine serum (FBS), penicillin/streptomycin (P/S), and phosphate-buffered saline (PBS) were purchased from Gibco-Invitrogen (Paisley, UK). Cell counting kit 8 (CCK8) was purchased from MCE (MedChemExpress LLC, Monmouth Junction, NJ, USA). Griess reagent, fluorescein isothiocyanate-dextran (FITC-Dextran), and lipopolysaccharides (LPS) were purchased from Sigma-Aldrich (St. Louis, MO, USA). An endotoxin detection-specific Limulus test kit was purchased from Bioendo Technology (Xiamen, China). Trifluoroacetic acid (TFA) was purchased from Fisher Scientific (Thermo Fisher Scientific, Waltham, MA, USA). Nylon membrane filters (0.22/0.45 μm) were purchased from Millipore (Billerica, MA, USA). Deionized water was prepared using a Millipore MilliQ-Plus system (Millipore, Billerica, MA, USA). All the other reagents were of analytical grade.

### 2.2. Preparation of Polysaccharides

All samples with sticky nature were stored in a −80 °C refrigerator for 24 h, mixed with diatomaceous earth of equal amounts to avoid clogging during grounding, and sieved to a fine powder (250 μm ± 9.9 μm). Each sample (1.0 g) was immersed in 20.0 mL of 80% methanol, then refluxed in a Syncore parallel reactor (Büchi, Flawil, Switzerland) for 2 h at 65 °C. Subsequently, the extract solution was centrifuged at 4000 × *g* for 10 min (Allegra X-15 centrifuge; Beckman Coulter, Fullerton, CA, USA), and the supernatant was collected. The extracted residue was dried above a hot water bath (WB22 Memmert, Memmert Company, Schwabach, Germany). Then a microwave-assisted extractor (Multiwave 3000, Anton Paar GmbH, Graz, Austria) was used to extract polysaccharides from samples. The low molecular weight compounds of crude polysaccharides were removed using ultra centrifugal filters (molecular weight cut off 1 kDa, Pall Corporation, Port Washington, NY, USA) by centrifugation at 2500× *g*. After ultrafiltration, the fraction with a molecular weight of more than 1 kDa polysaccharides was obtained and stored in a 4 °C refrigerator for subsequent analysis after freeze-drying.

### 2.3. HPSEC-MALLS-RID Analysis

HPSEC-MALLS-RID was employed to detect the fractions with molecular weight and content of polysaccharides according to our previous report [21]. HPSEC-MALLS-RID was composed of an Agilent 1260 series LC/DAD system (Agilent Technologies, Palo Alto, CA, USA), multi-angle light scattering detectors (MALLS, Wyatt Technology Co., Santa Barbara, CA, USA), and a refractometer (RID, Optilab rEX, Wyatt Technology Co.) in series at 35 °C. The chromatographic column was TOSOH gel columns TSKgel G2500 PWXL (300 mm × 7.8 mm), and the mobile phase was 0.9% NaCl solution. The flow rate was 0.5 mL/min with a 100 μL injection volume. ASTRA 7.3.2 software (Wyatt technology Co., CA, USA) was used to process the data. 

### 2.4. HPAEC-PAD Analysis

HPAEC-PAD (Thermo Scientific™ Dionex™ ICS-5000^+^, Dionex, MA, USA) was used for the analysis of compositional monosaccharides according to a previous method with minor modification [22]. The sample solution was mixed with an equal volume of 2.0 mol/L TFA for complete acid hydrolysis at 80 °C for 3 h to gather complete acid hydrolysates (CAH). All the samples were filtered through a 0.45 μm membrane before analysis. The mobile phase consisted of 88% deionized water and 12% 10 mM NaOH, running for 22 min at a flow rate of 0.4 mL/min on a CarboPac PA200 (3 mm × 250 mm) (Thermo Scientific™ Dionex™ CarboPac™ PA200, Dionex, MA, USA) analytical column with a system temperature of 25 °C. Ten monosaccharide standards, including Fuc, Ara, Rha, Gal, Glc, Xyl, Man, Fru, GalA, and GlcA, were used to calculate the content of each monosaccharide in the samples.

### 2.5. ABTS Radical Scavenging Test

The assay of ABTS radical scavenging was slightly modified according to the method of Chen et al. [23]. In brief, 7 mmol/L ABTS aqueous solution and 2.5 mmol/L potassium persulfate aqueous solution were mixed in a ratio of 1:1, and then stood in dark for 12 h. This solution was diluted with deionized water to reach a 0.7 ± 0.05 absorbance value at 734 nm, and obtained the ABTS working solution. Sample solutions (0.5, 1, 2, 4, 8 mg/mL) were prepared. In a 96-well plate, 200 μL of ABTS working solution and 10 μL of the sample solution were added to each well, and the reaction was kept in the dark for 6 min. After the reaction, the absorbance was measured, and the ABTS clearance rate formula was as followed:(1)$$C\left(\%\right)=\left[1-\left(A_{1}-A_{2}\right)/A_{0}\right]\times 100$$
where *C* is the clearance rate, *A*_0_ is the control absorbance, *A*_1_ is the sample absorbance, and *A*_2_ is the background absorbance, which is to eliminate the interference of the tested solution.

### 2.6. Cell Culture

RAW 264.7 cells were purchased from American Type Culture Collection (ATCC, Rockville, MD, USA), and used for evaluating the effects of polysaccharides. Cells were cultured in DMEM supplemented with 10% FBS and 1% P/S at 37 °C in a humidified atmosphere of 5% CO_2_. 

### 2.7. Cytotoxicity Assay

RAW 264.7 cells (5 × 10^3^ cells/well) were cultured in 96-well microplates overnight, then treated with LPS (0.4 μg/mL) and a series of concentrations of sample polysaccharides for 24 h, respectively. An equal volume of culture medium was used as blank control. Subsequently, the original culture medium was discarded and stained with 100 μL of culture medium containing 10% CCK8 for 2 h in dark. The absorbance values were read at 450 nm, and the cell viability was calculated as the ratio of absorbance values between the sample and vehicle control group.

### 2.8. Nitric Oxide (NO) Determination

RAW 264.7 cells (5 × 10^4^ cells/well) were seeded in 96-well microplates overnight, and then cells were treated with a series of concentrations of sample polysaccharides and LPS (0.4 μg/mL) for 24 h, respectively. An equal volume of culture medium was used as vehicle control. Subsequently, 75 μL of supernatants were collected and mixed with an equal volume of modified Griess reagent at room temperature for 15 min. The absorbance was measured at 540 nm. NO production was expressed as the ratio of absorbance values between the sample and the LPS-treated group.

### 2.9. Phagocytic Activity Test

RAW 264.7 cells (5 × 10^4^ cells/well) were cultured in 96-well plates overnight, and then incubated with culture medium, LPS (0.4 μg/mL), and a series of concentrations of samples polysaccharides for 18 h, respectively. Then, the cells were treated with FITC-dextran (0.1 mg/mL in culture medium) and incubated at 37 °C for an additional 1 h in dark. After incubation, the cells were collected with cold PBS after being washed three times. A BD Accuri™ C6 Cytometer (BD Biosciences, San Jose, CA, USA) was used to analyze. The percentage of phagocytosis was expressed as the ratio of phagocytic rate between treatment and control cells.

### 2.10. Determination of Endotoxin Contamination

An endotoxin detection-specific Limulus test kit was used for avoiding endotoxin contamination. The results indicated that endotoxin contamination in the tested sample could be excluded.

### 2.11. Statistical Analysis

GraphPad Prism 8.0.2 (Dotmatics, San Diego, CA, USA) was used to analyze and process the data. Data were presented as mean ± SEM from at least three independent experiments for each sample. Statistical significance between the experimental groups was determined by Student’s *t*-test, and *p* values less than 0.05 were considered statistically significant. SPSS statistics 26 (IBM®, NY, USA) was used for Pearson correlation analysis. Origin Pro 2022b (OriginLab, MA, USA) was used for cluster analysis.

## 3. Results and Discussion

### 3.1. Optimization of Microwave-Assisted Extraction of Polysaccharides

Microwave-assisted extraction parameters were optimized by using sample OJZ7. Single-factor investigation was carried out on liquid-solid ratio, extraction power and time. All sample extracts were made up to 100 mL in volumetric flasks and filtered through a 0.45 μm filter membrane before analysis. HPSEC-MALLS-UV-RID profile of OJZ7 polysaccharides mainly had two peaks, Peak1 and Peak2, based on RI detection (Figure 1d). The effects of the three parameters on microwave-assisted extraction efficiency are also shown in Figure 1. The results indicated that microwave power had a great effect on polysaccharide extraction, and the contents of polysaccharides extracted at both 300 W and 100 W were similar, but the content rapidly decreased as the power increased to 500 W. The optimum extraction time and liquid-solid ratio were 15 min and 40, respectively (Figure 1b,c). From the optimization results of MAE, the polysaccharide content significantly decreased as the extraction power further increased, possibly due to the damage caused by the higher power on the polysaccharide structure (Figure 1a). The low polysaccharide content was seen at 10 min of extraction time, while as time increased, the poly-saccharides content increased at 15 min and decreased again at 20 min, which could be due to the excessive time that would lead to polysaccharide cleavage (Figure 1b). The polysaccharide content reached the maximum at a liquid-solid ratio of 40. This may be because as the liquid-solid ratio increases, more polysaccharides can be extracted. Then, due to solvent saturation, the increased liquid-solid ratio made it more difficult for microwave action on the samples (Figure 1c). Finally, to ensure stable extraction efficiency, we extracted OJZP, OJCP, LMP, and LSP, polysaccharides from 29 batches of samples under the power of 300 W, by using MAE with a liquid-solid ratio of 40 for 15 min.

### 3.2. Molecular Parameters of MaiDong Polysaccharides

All the sample polysaccharides were dissolved in HPSEC mobile phase at 2 mg/mL and filtered through a 0.45 μm filter membrane before analysis. HPSEC-MALLS-UV-RID chromatograms of representative samples are shown in Figure 2. Each was divided into three peaks based on MALLS, UV, and RI detection. Molecular parameters of the three peaks in all samples are listed in Table 2. The results indicated that in all samples, HPSEC profiles were very similar, but their molecular parameters, including Mw, could be variously different. Briefly, all samples had similar laser and UV signal profiles, but there were differences in the values, and the polysaccharides from different sources of *MaiDong* had some differences in the RID signal profiles; however, the RID signal profiles were similar when compared within the same source samples. Furthermore, all polysaccharide samples from fibrous roots had higher RID response values at Peak2 than those from the root. The Mw analysis revealed that OJZP, OJCP, and LMP had similar molecular weight distributions, with LSP having a larger molecular weight than LMP. Meanwhile, polysaccharides in OJZ8, OJZ9, and OJZ10 had higher molecular weights than other OJZP. However, polysaccharides in OJC8, extracted from the fibrous root, did not significantly differ in terms of molecular weight from other OJCP. Similarly, growth years had no consistent effect on polysaccharide molecular weight. The molecular weight of the polysaccharides, the RID signal, the laser signal, and the UV signal were all considered for inclusion in the quality control indicator, after combining the above information.

### 3.3. Monosaccharide Composition

After complete acid hydrolysis, monosaccharide compositions of polysaccharides were determined by HPAEC-PAD. Table 3 shows their compositional monosaccharides molar ratios. From the monosaccharide composition results, all polysaccharides were mainly composed of Fru and Glc, with small amounts of Ara, Man, Gal, and Xyl. The molar ratio of Fru to Glc was approximately 15:1 in OJZP, 14:1 in OJCP, 11:1 in LMP, and 13:1 in LSP, with relatively similar results, implying that the different sources of polysaccharides in *MaiDong* have a consistent monosaccharide composition, but with minor differences in content, which was similar to other research [24,25]. Normally, the fibrous roots of *MaiDong* are removed for medicinal purposes, but the results of this study suggested that the polysaccharides in the fibrous roots may be complementary to those in the root, to some extent. At the same time, because the growth years did not result in significant changes in monosaccharide composition, it may be possible to shorten the growth years to save cultivation costs.

### 3.4. ABTS Free Radical Scavenging Ability of MaiDong Polysaccharides

*MaiDong* was thought to have antioxidant properties [6]. Modern research has demonstrated that oxygen-containing radicals in the human body oxidize cell membranes, accelerate aging, and damage DNA [26]. ABTS radicals are a common form of oxygen-containing radical, and drugs’ ability to scavenge them can, to some extent, reflect their antioxidant capacity [27]. In this study, we investigated and compared the ability of various polysaccharides from *MaiDong* to scavenge ABTS radicals. Figure 3 depicts the antioxidant capacity of all polysaccharides from *MaiDong*, and their 20% maximal inhibitory concentration (IC_20_) for ABTS scavenging ability is listed in Table 4. All samples demonstrated the ability to scavenge ABTS radicals to varying degrees. According to Table 4, polysaccharides extracted from fibrous roots were more effective at ABTS scavenging than those extracted from the root. Polysaccharides from OJZ8, OJZ9, OJZ10, and OJC8 had IC_20_ values of 1.81, 1.87, 1.40, and 2.30 mg/mL, respectively, whereas the IC_20_ values of the polysaccharides extracted from the root were all greater than 3 mg/mL. The fibrous roots have a substantially lower molar ratio of Fru and Ara in *Zhe MaiDong* and *Chuan MaiDong* than that of their tuberous roots, which might contribute to the varied antioxidant activity. In general, the antioxidant capacity of OJZP was comparable to that of OJCP in terms of ABTS scavenging rate and IC_20_ value, and both were much higher than that of LMP and LSP, with LMP slightly higher than LSP. It is interesting to note that the antioxidant capacity of OJZP did not change consistently over the growth years, according to the results of the ABTS radical scavenging test (Figure 3 and Table 4). The ratio of various monosaccharides in the composition and how glycosidic bonds are connected, in our opinion, may be suspect for the variations in the antioxidant capacity of polysaccharides. Furthermore, different molecular weights and contents may have an effect on the results. Based on the antioxidant capacity of polysaccharides from *MaiDong*, the ABTS radical scavenging capacity should be used as an indicator for evaluating the quality of polysaccharides in different origins of *MaiDong*.

### 3.5. Immunostimulatory Activity of OJZP, OJCP, LMP and LSP

Macrophages play an indispensable role in the innate and adaptive immunity of the human body [28]. Studies have shown that high levels of NO are associated with immune responses during antitumor and antiviral processes, which can trigger cell proliferation, apoptosis, signal transduction, immune defense, and other physiological processes [29]. Phagocytosis is a basic cellular process that plays an important role in the immune system [30]. In this study, RAW 264.7 cells were treated with a series of concentrations of polysaccharides from selected samples and LSP, and their effects on NO production and phagocytic activity were investigated.

We selected a series of sample concentrations to determine the effect of cell viability on the results of subsequent experiments, and finally chose concentrations of 5, 10, and 20 μg/mL. All samples in this series neither significantly promoted RAW 264.7 cell proliferation nor had cytotoxicity; the results are shown in Figure 4a. Effects of these polysaccharides on NO production of macrophages are shown in Figure 4b. The results of NO release showed that, among the polysaccharides extracted from the root of *O*. *japonicus* in Zhejiang with different growth years, all samples had a strong ability to promote NO release except OJZ13. Meanwhile, among the polysaccharides extracted from the root of *O*. *japonicus* in Sichuan, *L*. *muscari* and *L*. *spicata* with one-year-growth, not all samples promoted NO production, and showed no significant difference from the control group at the concentrations of 5, 10, and 20 μg/mL, except for OJC3. OJZ9 and OJC8 were both polysaccharides extracted from fibrous roots, and they were found to be more effective at promoting NO release than polysaccharides from the root. It has been reported that polysaccharides rich in β-1,4-D-Man*p* may help to activate the immune system by attaching to the mannose receptor on macrophages [31]. As a result, the varied monosaccharide composition ratios, particularly the variable Man molar ratios, are likely related to the immunological activities of OJZ9 and OJC8 on RAW 264.7 cells.

The results of phagocytosis experiments of polysaccharides from *MaiDong* were consistent with the results of NO release, reflecting the following overall trends (Figure 4c). First, the polysaccharides extracted from fibrous roots (OJZ9 and OJC8) had a stronger ability to promote FITC-Dextran phagocytosis by macrophages than that of the root. Second, OJZP had a stronger ability to promote phagocytosis than OJCP, LMP, and LSP. Third, LMP and LSP almost did not promote phagocytosis at the concentrations of 5, 10, and 20 μg/mL. Fourth, the macrophage-promoting ability of 3-year-old OJZ11 was stronger than that of 2-year-old OJZ12, but no significant difference from that of 1-year-old OJZP16, so it was not possible to determine whether the growth age had a significant effect on the phagocytosis of polysaccharides from *MaiDong*, and more samples are needed for further investigation; though *MaiDong* produced in Zhejiang should be collected at the third year. As can be seen in the typical phagocytosis flow charts (Appendix A), the majority of the OJZP had a significant right shift compared to the control group, with the exception of the OJZ6 and OJZ13. This suggested that the stimulation of macrophages with OJZP could promote their phagocytosis to a certain extent. In contrast, as shown in the flow diagrams of samples from various origins, except for OJC3 and OJC8, the peaks of all samples showed no right shift when compared to the control group, which implied that the effect of OJCP, LMP, and LSP on enhancing the phagocytic ability of macrophages is not as significant as that of OJZP. In summary, the experimental results revealed that polysaccharides from fibrous roots promoted NO production (Figure 4b) and phagocytosis (Figure 4c) more effectively than those from the root. Additionally, OJZP promoted NO production more effectively than OJCP, and more strongly than LMP and LSP.

### 3.6. Multivariate Statistical Analysis for Polysaccharide Quality Control in MaiDong

#### 3.6.1. Relationship of Chemicals and Activity

Scientific research has revealed that the biological activity of polysaccharides is frequently linked to their structure [32]. Using Pearson correlation analysis, we investigated the relationship of polysaccharide physicochemical properties with pharmacological activities, such as ABTS radical scavenging ability and immunological activity. 

Pearson correlation analysis is used to reflect the linear correlation of variables, and the Pearson correlation coefficient is a measurement of the linear correlation between two sets of data [33]. A Pearson correlation coefficient that shows significance indicates that there is a correlation between the data. After that, the absolute value of the magnitude of the Pearson correlation coefficient reflects the closeness of the relationship between the data. A coefficient between 0.8 to 1 indicates a very close relationship; between 0.3 and 0.8 indicates a close relationship; and between 0 to 0.3 indicates a low correlation. 

The correlation coefficients between the pharmacological activity and physicochemical properties of polysaccharides in *MaiDong* are recorded in Appendix A. In brief, Pearson correlation analysis was performed on 29 samples using ABTS radical scavenging ability as the reference, and 14 samples using immune activity as the reference. The comparison indicators were monosaccharide composition, Mw, RI, UV, and laser signal. In addition, the correlation values are calculated and recorded in Table 5. The correlation values ranged from 0 to 1. According to the table, the content of Man had the highest correlation value of 0.904 for antioxidant capacity among the 15 evaluated indicators, followed by the content of Xyl, with 0.896. Fru, the richest monosaccharide in *MaiDong* polysaccharides, had a significantly negative relationship to antioxidant and immunological activity of the polysaccharides. Meanwhile, the Mw at Peak1 and UV at Peak2 reflected no correlation with antioxidant activity. For immune activity, the content of Man also had the highest correlation value 0.817, followed by that of Ara at 0.713, while the UV signal at Peak2, and Mw at both Peak1 and Peak2, did not show a correlation with immune activity. 

#### 3.6.2. Quality Marker Optimization

In addition to the molecular parameters, monosaccharide composition, and pharmacological activity of polysaccharides, the yield of polysaccharides is also an important factor to evaluate their quality [34]. The higher yield of polysaccharide means that the cost of planting can be reduced, which is beneficial to the marketization of *MaiDong*. In conclusion, seven indicators were chosen for polysaccharide quality assessment: yield, Mw, RI, laser, UV, monosaccharide composition, and ABTS scavenging activity.

Cluster analysis, by considering the seven indicators together, results in a spectrum plot as shown in Figure 5a. Such a clustering method can distinguish fibrous roots from the root and distinguish OJZP from OJCP. However, it does not distinguish OJCP from LMP and LSP, nor can it distinguish OJZP by growth years. Considering that, in general, polysaccharides do not have UV signals, we excluded the UV indicator. The results after optimization of the indexes are shown in Figure 5b. The results showed that by using the indicators of yield, ABTS clearance capacity, and the content of fructose, RID signal intensity could most effectively distinguish *O*. *japonicus* in Zhejiang from other *MaiDong* and could distinguish the difference between fibrous roots and root. *L*. *muscari* and *L*. *spicata* were also distinguished from *O*. *japonicus* in Sichuan. However, the growing years did not show an influence on the results. MaiDong produced in Sichuan usually can be collected in the first year, but *MaiDong* produced in Zhejiang should be collected in the third year. The difference might be mainly derived from the climate, and cultivation conditions should also be carefully investigated in the future. Nevertheless, chemical variation [11,14,15,16] has been noticed and their clinical efficacy should be carefully compared.

Through the Pearson correlation analysis, we found that Man, Ara, and Xyl showed significantly high correlation values for antioxidant capacity and immune activity, while the signal of UV and Mw showed low correlation values. It has been shown that the structure of mannose backbone or side chain modifications in polysaccharides has a relevant link to their mediated biological activities, such as immunomodulatory activities [35]. This may be one of the reasons why the Man content reflects a strong correlation in this analysis. As for how these physicochemical property indicators affect pharmacological activity, we believe that further in-depth exploration is needed through subsequent experiments.

Finally, through the cluster analysis (Figure 5), the yield, content of Fru, ABTS scavenging activity, and the RI signal were used as indicators for clustering. Through the results, a clear distinction can be made between the OJZ, OJC, LS, and LM, and the fibrous roots and the root. This may provide an innovative, simplified, and prompt strategy for the quality control of *MaiDong*. However, most likely due to the nature of the samples themselves, the properties of polysaccharides in *L*. *muscari*, *L*. *spicate*, and *O*. *japonicus* in Sichuan were closer.

## 4. Conclusions

In this study, the physicochemical properties and pharmacological activities of different sources of *MaiDong* were comprehensively compared, and the results showed that different *MaiDong* had similar molecular parameters and compositional monosaccharides. *O*. *japonicus* in Zhejiang had better antioxidant and immune-promoting activities than *O*. *japonicus* in Sichuan. There was no effect of growing age on the pharmacological activity of *O*. *japonicus* in Zhejiang. The fibrous root had better pharmacological activity than the root. The polysaccharides yield, content of Fru, antioxidant activity, and RI signal can be employed as quality indicators of *MaiDong*, which is helpful to the cultivation, selection, use, and quality control of *MaiDong*.

## Figures and Tables

**Figure 1 biomolecules-12-01491-f001:**
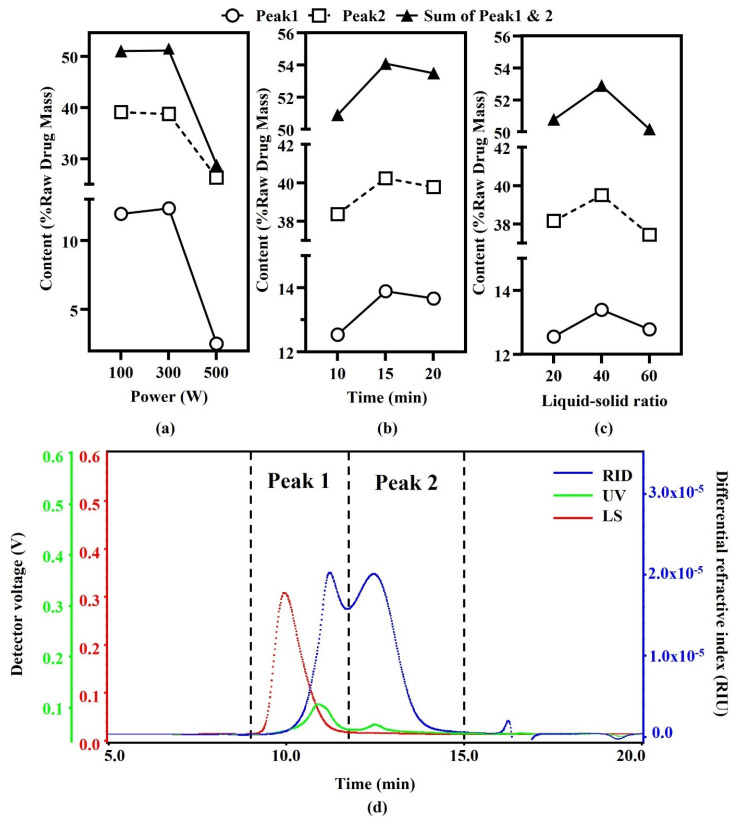
Effect of microwave power (**a**), extraction time (**b**), and liquid-solid ratio (**c**) on extraction efficiency of polysaccharides from *Ophiopogon japonicus* in Zhejiang based on (**d**) HPSEC-MALLS-RID chromatogram with peaks 1 and 2.

**Figure 2 biomolecules-12-01491-f002:**
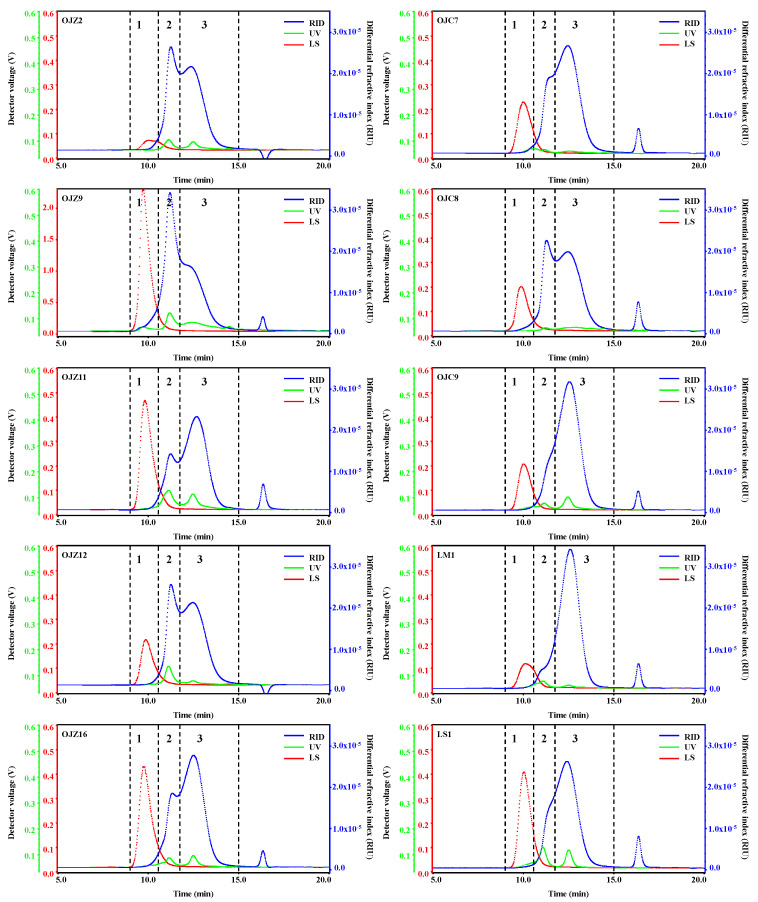
Representative HPSEC-MALLS-RID chromatograms with peaks 1, 2 and 3 of polysaccharides from different samples. The codes are the same as in Table 1.

**Figure 3 biomolecules-12-01491-f003:**
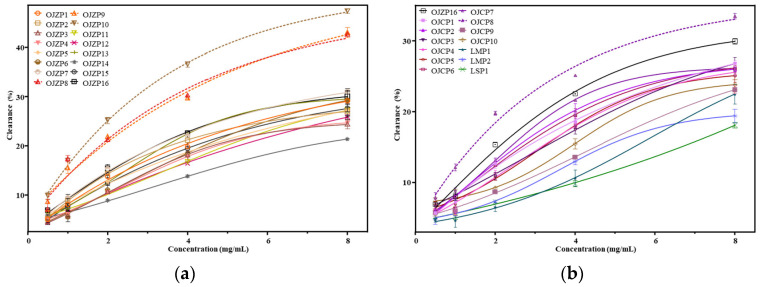
ABTS free radical scavenging activity of polysaccharides from *MaiDong* with different (**a**) and same (**b**) growth years. The codes are the same as in Table 1.

**Figure 4 biomolecules-12-01491-f004:**
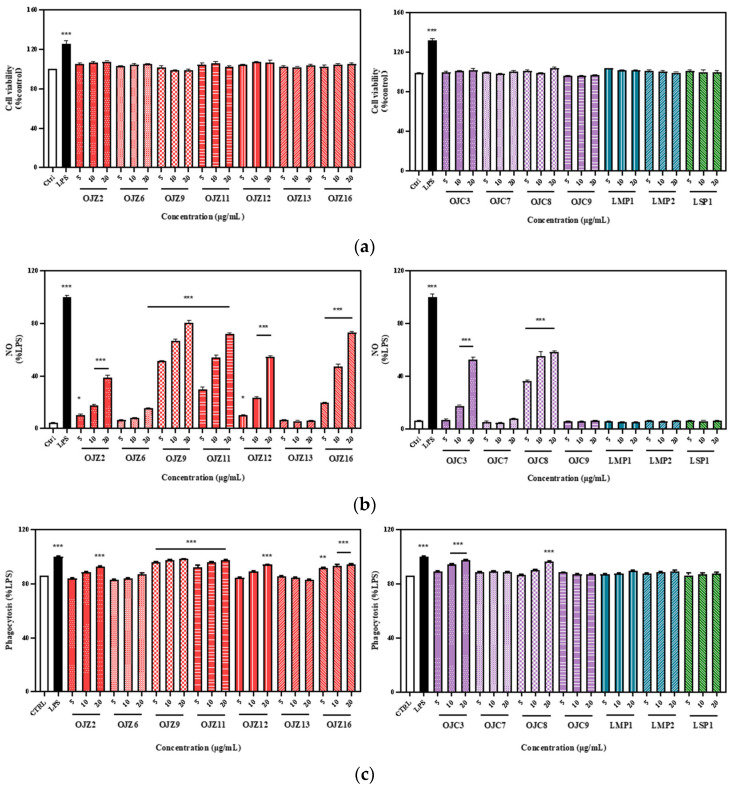
Effects of polysaccharides from representative samples of *MaiDong* on cell viability (**a**), NO production (**b**), and phagocytosis (**c**) of RAW 264.7 macrophages. All values were expressed as mean ± SEM of three independent experiments. * *p* < 0.05, ** *p* < 0.01, *** *p* < 0.001 vs control group, unmarked results indicate no significant difference vs control group. The codes are the same as in Table 1.

**Figure 5 biomolecules-12-01491-f005:**
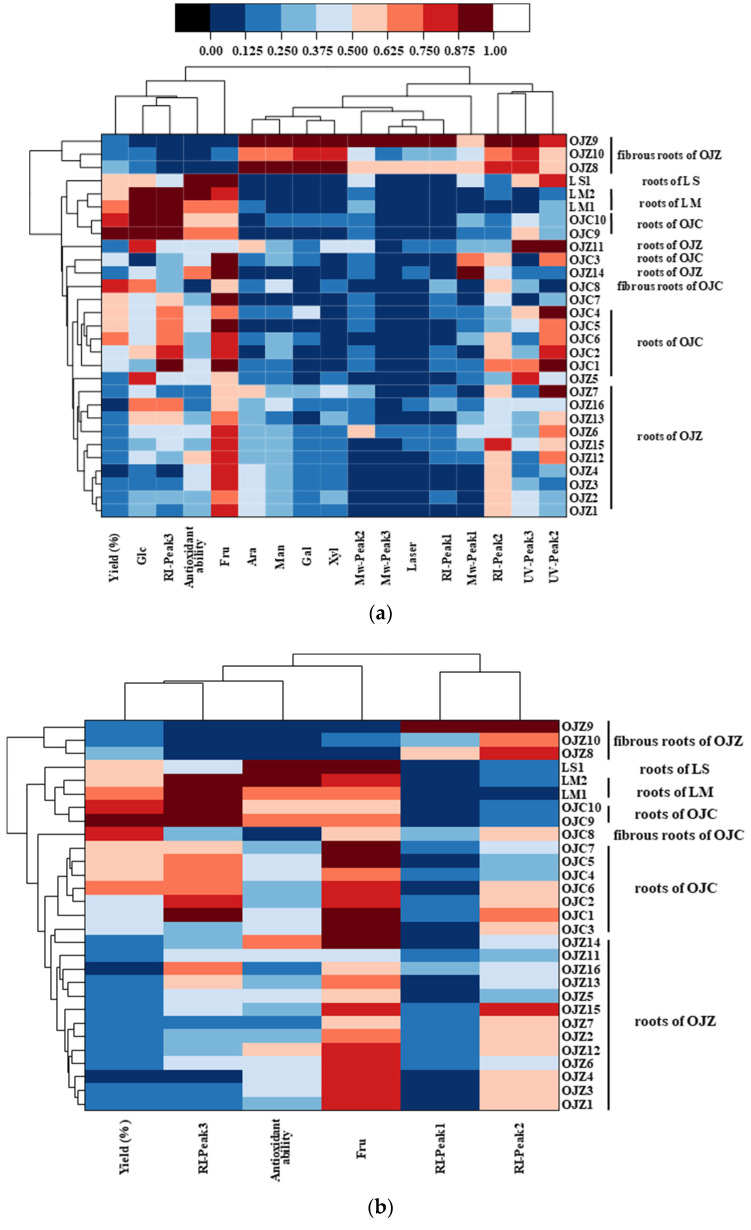
Hierarchical clustering and heat-map of 29 batches of *MaiDong* polysaccharides based on (**a**) all investigated indicators or (**b**) optimized indicators.

**Table 1 biomolecules-12-01491-t001:** The detailed information of 29 samples of *Maidong*.

Species	Sample ID	Origin	Plant part	Harvest Time	Growth Years
*Ophiopogon japonicus*	OJZ1 ^1^	Zhejiang	Root	17 May 2020	3
	OJZ2	Zhejiang	Root	9 May 2020	3
	OJZ3	Zhejiang	Root	1 June 2020	3
	OJZ4	Zhejiang	Root	9 May 2020	3
	OJZ5	Zhejiang	Root	17 May 2020	3
	OJZ6	Zhejiang	Root	17 May 2021	3
	OJZ7	Zhejiang	Root	23 June 2021	3
	OJZ8	Zhejiang	Fibrous Roots	17 May 2020	3
	OJZ9	Zhejiang	Fibrous Roots	9 May 2020	3
	OJZ10	Zhejiang	Fibrous Roots	1 June 2020	3
	OJZ11	Zhejiang	Root	August 2020	3
	OJZ12	Zhejiang	Root	August 2020	2
	OJZ13	Zhejiang	Root	August 2020	2
	OJZ14	Zhejiang	Root	August 2020	2
	OJZ15	Zhejiang	Root	August 2020	2
	OJZ16	Zhejiang	Root	August 2020	1
	OJC1 ^2^	Sichuan	Root	July 2020	1
	OJC2	Sichuan	Root	July 2020	1
	OJC3	Sichuan	Root	July 2020	1
	OJC4	Sichuan	Root	July 2020	1
	OJC5	Sichuan	Root	July 2020	1
	OJC6	Sichuan	Root	July 2020	1
	OJC7	Sichuan	Root	July 2020	1
	OJC8	Sichuan	Fibrous Roots	July 2020	1
	OJC9	Sichuan	Root	August 2021	1
	OJC10	Sichuan	Root	August 2021	1
*Liriope muscari*	LM1 ^3^	Fujian	Root	August 2021	1
	LM2	Fujian	Root	August 2021	1
*Liriope spicata*	LS1 ^4^	Hubei	Root	August 2021	1

^1^ Polysaccharides from *O*. *japonicus* in Zhejiang; ^2^ Polysaccharides from *O*. *japonicus* in Sichuan; ^3^ Polysaccharides from *L. muscari*; ^4^ Polysaccharides from *L*. *spicata*.

**Table 2 biomolecules-12-01491-t002:** The molecular weight (Mw), polydispersity index (Mw/Mn), z-average radius of gyration (Rz), content of different peak (fraction), and yield of polysaccharides from *MaiDong* of different species or locations.

Sample ID	Peak 1				Peak 2				Peak 3				Recovery (%)
	Mw (×10^6^ Da)	Mw/Mn	Rz (nm)	Content (%)	Mw (×10^4^ Da)	Mw/Mn	Rz (nm)	Content (%)	Mw (×10^3^ Da)	Mw/Mn	Rz (nm)	Content (%)	
OJZ1	7.2 (±2.8%)	2.9 (±3.3%)	96.5 (±1.9%)	0.7	6.0 (±1.1%)	3.4 (±1.6%)	25.3 (±6.3%)	32.3	6.0 (±1.8%)	1.0 (±2.5%)	25.6 (±10.1%)	40.2	7.3
OJZ2	6.7 (±2.3%)	2.7 (±3.3%)	93.4 (±1.6%)	0.9	6.8 (±0.9%)	3.7 (±1.4%)	25.1 (±5.1%)	35.1	6.0 (±1.6%)	1.0 (±2.2%)	23.1 (±10.5%)	42.2	7.4
OJZ3	12 (±2.8%)	3.8 (±3.1%)	86.0 (±1.9%)	0.6	6.5 (±1.3%)	3.2 (±1.9%)	32.0 (±4.6%)	32.0	7.9 (±1.9%)	1.0 (±2.6%)	39.3 (±4.8%)	40.5	6.9
OJZ4	6.8 (±2.2%)	3.1 (±2.5%)	77.7 (±1.7%)	0.6	5.9 (±1.1%)	3.2 (±1.7%)	24.2 (±7.1%)	32.7	6.2 (±1.9%)	1.0 (±2.6%)	23.2 (±12.6%)	36.8	6.0
OJZ5	61 (±7.6%)	5.6 (±7.7%)	158.1 (±2.3%)	0.6	13 (±1.1%)	4.3 (±2.1%)	41.4 (±2.5%)	26.3	9.1 (±2.9%)	1.1 (±4.0%)	60.4 (±3.3%)	47.4	7.9
OJZ6	170 (±9.5%)	4.9 (±9.8%)	201.4 (±1.4%)	1.1	43 (±3.4%)	2.7 (±5.8%)	85.6 (±2.2%)	30.2	77 (±4.1%)	1.2 (±5.9%)	147.7 (±1.2%)	47.8	8.1
OJZ7	48 (±6.0%)	4.2 (±6.2%)	141.3 (±1.9%)	1.2	24 (±1.4%)	4.0 (±2.7%)	42.4 (±2.6%)	32.0	22 (±3.7%)	1.1 (±5.1%)	81.4 (±2.2%)	41.2	8.6
OJZ8	190 (±13.1%)	8.1 (±13.3%)	204.1 (±1.5%)	2.2	44 (±4.1%)	2.1 (±7.0%)	102.2 (±2.0%)	40.4	220 (±5.3%)	1.2 (±7.5%)	223.2 (±1.2%)	32.8	10.3
OJZ9	180 (±12.4%)	5.3 (±12.9%)	209.8 (±1.4%)	4.0	71 (±5.4%)	2.0 (±8.6%)	116.2 (±1.8%)	47.6	390 (±6.4%)	1.1 (±9.0%)	229.7 (±1.1%)	34.3	7.2
OJZ10	170 (±11.8%)	7.3 (±12.0%)	192.5 (±1.7%)	1.7	31 (±3.1%)	2.6 (±5.6%)	83.5 (±1.8%)	37.3	92 (±4.8%)	1.2 (±6.8%)	169.8 (±1.2%)	34.7	7.3
OJZ11	120 (±12.3%)	8.0 (±12.4%)	184.7 (±2.5%)	1.0	32 (±1.9%)	3.2 (±4.6%)	55.9 (±2.1%)	23.1	28 (±5.2%)	1.2 (±7.4%)	123.2 (±1.7%)	49.9	8.2
OJZ12	52 (±9.0%)	5.8 (±9.2%)	150.3 (±2.8%)	0.8	10 (±1.5%)	3.9 (±2.4%)	42.9 (±2.9%)	33.5	9.9 (±3.3%)	1.1 (±4.5%)	61.9 (±3.5%)	44.7	8.0
OJZ13	130 (±8.6%)	4.7 (±8.8%)	145.7 (±2.3%)	0.7	21 (±1.4%)	4.4 (±2.6%)	51.6 (±1.8%)	30.3	18 (±3.2%)	1.1 (±4.4%)	86.5 (±1.7%)	53.2	7.0
OJZ14	340 (±26.8%)	4.5 (±27.0%)	232.5 (±3.5%)	0.4	18 (±2.3%)	4.2 (±4.4%)	65.9 (±1.8%)	29.2	25 (±5.2%)	1.3 (±7.1%)	120.4 (±1.6%)	45.4	6.8
OJZ15	100 (±17.9%)	10.3 (±18.0%)	208.7 (±2.7%)	1.1	13 (±2.4%)	3.0 (±4.6%)	58.0 (±2.4%)	39.9	28 (±5.1%)	1.2 (±7.3%)	119.6 (±1.8%)	46.9	7.4
OJZ16	87 (±11.3%)	9.1 (±11.4%)	185.4 (±2.0%)	1.3	20 (±2.3%)	3.4 (±5.3%)	57.3 (±2.4%)	29.8	23 (±6.3%)	1.2 (±8.7%)	112.6 (±2.2%)	55.1	4.4
Mean	105	5.6	160.5	1.2	22.8	3.3	56.8	33.2	60.5	1.1	103	43.3	7.4
OJC1	62 (±7.4%)	5.3 (±7.5%)	156.8 (±2.0%)	1.0	16 (±1.5%)	5.0 (±3.2%)	43.4 (±2.5%)	35.9	12 (±4.7%)	1.1 (±6.4%)	74.2 (±3.1%)	68.0	12.8
OJC2	44 (±4.3%)	4.1 (±4.5%)	139.6 (±1.4%)	0.8	15 (±1.2%)	6.6 (±2.4%)	35.5 (±2.9%)	32.8	9.1 (±3.3%)	1.1 (±4.3%)	54.6 (±3.8%)	60.0	12.9
OJC3	240 (±9.4%)	3.6 (±9.6%)	151.7 (±2.3%)	0.3	20 (±1.5%)	3.9 (±2.8%)	54.6 (±1.7%)	33.6	27 (±2.6%)	1.2 (±3.6%)	88.3 (±1.3%)	45.2	13.1
OJC4	65 (±8.9%)	5.1 (±9.0%)	156.1 (±2.6%)	0.8	22 (±1.4%)	6.3 (±2.7%)	40.5 (±2.8%)	26.1	11 (±3.2%)	1.1 (±4.2%)	66.6 (±2.6%)	55.9	13.7
OJC5	88 (±7.8%)	4.0 (±8.0%)	139.6 (±2.7%)	0.7	22 (±1.6%)	6.8 (±2.9%)	46.3 (±2.4%)	24.1	12 (±3.3%)	1.1 (±4.4%)	72.1 (±2.3%)	57.3	14.7
OJC6	110 (±12.7%)	7.6 (±12.9%)	161.9 (±2.9%)	0.7	13 (±2.0%)	3.9 (±3.7%)	53.9 (±2.3%)	34.3	17 (±4.1%)	1.2 (±5.6%)	89.0 (±2.1%)	56.2	15.3
OJC7	46 (±6.0%)	3.4 (±6.2%)	119.7 (±2.7%)	0.8	16 (±1.5%)	7.5 (±2.4%)	38.8 (±3.3%)	28.2	8.1 (±2.4%)	1.1 (±3.3%)	48.7 (±3.5%)	54.5	19.0
OJC8	41 (±8.8%)	4.8 (±8.9%)	194.1 (±2.3%)	1.3	11 (±2.2%)	3.2 (±4.5%)	59.1 (±2.1%)	31.0	18 (±5.2%)	1.1 (±7.3%)	97.2 (±2.3%)	45.1	13.6
OJC9	69 (±7.2%)	2.8 (±7.8%)	128.2 (±3.1%)	0.5	20 (±1.8%)	8.6 (±2.7%)	41.8 (±3.3%)	19.9	7.3 (±2.6%)	1.1 (±3.4%)	49.7 (±3.5%)	64.1	21.8
OJC10	100 (±9.3%)	4.3 (±9.6%)	142.6 (±3.7%)	0.3	19 (±1.9%)	6.5 (±3.0%)	45.8 (±3.0%)	20.5	8.0 (±2.8%)	1.1 (±3.8%)	51.2 (±3.5%)	67.4	18.7
Mean	86.5	4.5	149.0	0.7	13.4	5.8	46.0	28.6	12.9	1.1	69.2	57.4	15.6
LM1	37 (±5.3%)	2.4 (±5.7%)	116.1 (±2.3%)	0.5	29 (±1.5%)	9.5 (±2.5%)	35.4 (±3.9%)	14.4	6.8 (±2.6%)	1.1 (±3.4%)	44.3 (±4.4%)	64.0	15.5
LM2	36 (±3.3%)	1.7 (±3.7%)	88.0 (±2.0%)	0.3	17 (±1.4%)	7.7 (±2.2%)	36.3 (±3.6%)	20.6	7.0 (±2.1%)	1.1 (±2.9%)	39.6 (±4.7%)	65.6	14.6
Mean	36.5	2.0	102.1	0.4	23	8.6	35.8	17.5	6.9	1.1	41.9	64.8	15
LS1	150 (±9.9%)	2.4 (±10.1%)	139.4 (±3.4%)	0.4	36 (±1.5%)	8.4 (±2.5%)	49.1 (±2.1%)	22.4	17 (±2.6%)	1.1 (±3.7%)	78.1 (±1.8%)	50.0	14.6

The samples ID are the same as in Table 1.

**Table 3 biomolecules-12-01491-t003:** Molar ratio of compositional monosaccharides of OJZP, OJCP, LMP, and LSP.

Samples	Fru	Glc	Ara	Man	Gal	Xyl
OJZ1	87.73	5.57	1.00	0.94	0.61	0.31
OJZ2	79.27	5.05	1.00	0.84	0.63	0.33
OJZ3	87.75	5.47	1.00	0.80	0.57	0.28
OJZ4	90.44	5.46	1.00	0.77	0.63	0.30
OJZ5	122.24	10.35	1.00	0.79	0.59	0.32
OJZ6	108.21	7.35	1.00	0.90	0.64	0.32
OJZ7	68.83	4.74	1.00	0.67	0.65	0.37
OJZ8	43.26	2.70	1.00	1.19	1.30	0.53
OJZ9	39.72	2.28	1.00	1.09	1.35	0.49
OJZ10	56.36	3.53	1.00	1.13	1.49	0.55
OJZ11	66.73	5.61	1.00	0.73	0.52	0.39
OJZ12	116.18	8.01	1.00	0.94	0.66	0.44
OJZ13	110.26	8.21	1.00	0.62	0.56	0.44
OJZ14	205.93	14.16	1.00	0.38	0.64	0.34
OJZ15	114.89	7.34	1.00	1.03	0.63	0.41
OJZ16	100.61	7.89	1.00	1.23	0.63	0.38
Mean	93.65	6.48	1.00	0.88	0.76	0.39
SD	39.48	2.97	0.00	0.23	0.31	0.08
RSD	42.16	45.78	0.00	25.76	41.48	21.44
OJC1	164.22	10.83	1.00	1.09	0.87	0.32
OJC2	192.41	13.76	1.00	1.61	1.03	0.41
OJC3	146.07	8.08	1.00	1.22	0.83	0.24
OJC4	152.66	10.54	1.00	0.94	2.26	0.35
OJC5	218.65	15.29	1.00	0.94	1.15	0.37
OJC6	164.62	11.54	1.00	1.57	0.98	0.37
OJC7	185.86	12.92	1.00	0.79	0.79	0.27
OJC8	126.33	9.82	1.00	1.77	0.61	0.46
OJC9	219.37	19.59	1.00	0.99	1.11	0.44
OJC10	180.42	15.96	1.00	1.22	1.19	0.48
Mean	175.10	12.83	1.00	1.21	1.08	0.37
SD	30.28	3.42	0.00	0.33	0.45	0.08
RSD	17.29	26.62	0.00	27.30	41.73	21.36
LM1	161.76	14.41	1.00	0.35	0.58	0.21
LM2	344.84	29.55	1.00	0.95	0.58	0.44
Mean	253.30	21.98	1.00	0.65	0.58	0.33
SD	129.50	10.71	0.00	0.42	0.00	0.16
RSD	51.13	48.73	0.00	65.28	0.00	50.03
LS1	236.06	17.31	1.00	0.89	0.64	0.35

**Table 4 biomolecules-12-01491-t004:** ABTS free radical scavenging ability of *MaiDong* polysaccharides.

Sample	IC_20_ (mg/mL)
OJZP1	3.88
OJZP2	3.56
OJZP3	4.64
OJZP4	4.79
OJZP5	4.50
OJZP6	4.43
OJZP7	3.23
OJZP8	1.81
OJZP9	1.87
OJZP10	1.40
OJZP11	4.99
OJZP12	5.30
OJZP13	3.47
OJZP14	7.03
OJZP15	4.19
OJZP16	3.22
OJCP1	4.47
OJCP2	3.94
OJCP3	4.91
OJCP4	4.68
OJCP5	4.61
OJCP6	4.16
OJCP7	3.50
OJCP8	2.30
OJCP9	6.45
OJCP10	5.33
LMP1	7.10
LMP2	>IC_15_: 4.73
LSP1	>IC_15_: 6.66

**Table 5 biomolecules-12-01491-t005:** Pearson correlation analysis between activities and physicochemical characters.

Rank	Antioxidant Activity		Immune Activity	
	Indicator	Correlation Coefficient	Indicator	Correlation Coefficient
1	Man	0.904 **	Man	0.817 **
2	Xyl	0.896 **	Ara	0.713 **
3	Gal	0.863 **	Xyl	0.688 **
4	Ara	0.832 **	RI-Peak3	−0.657 *
5	RI-Peak1	0.815 **	RI-Peak2	0.626 *
6	Fru	−0.800 **	RI-Peak1	0.616 *
7	LS	0.753 **	Fru	−0.598 *
8	Mw-Peak3	0.708 **	Gal	0.567 *
9	RI-Peak2	0.686 **	UV-Peak3	0.509
10	RI-Peak3	−0.609 **	LS	0.491
11	UV-Peak3	0.515 **	Mw-Peak3	0.442
12	Glc	−0.510 **	Glc	−0.439
13	Mw-Peak2	0.507 **	UV-Peak2	0.341
14	Mw-Peak1	0.238	Mw-Peak1	0.229
15	UV-Peak2	0.151	Mw-Peak2	0.207

* *p* < 0.05, ** *p* < 0.01.

## Data Availability

Data will be provided upon reasonable request.

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
