# Peer review of "Quality Evaluation of Ophiopogon japonicus from Two Authentic Geographical Origins in China Based on Physicochemical and Pharmacological Properties of Their Polysaccharides"

_biomolecules, 2022, doi:10.3390/biom12101491_

Round 1

Reviewer 1 Report

In this manuscript, the authors utilized new evaluation method of polysaccharides and their pharmacological properties to qualify Ophiopogon japonicas. It is significant for the cultivation of Maidong. However, there were some problems to be solved as it followed.

1.       The Abstract should be revised carefully. In my opinion, it was not expressed the advantages of the new method and the conclusion. Some values should be added in order to show the results.

2.       In Introduction, some known method of evaluation should be introduced.

3.       Line 76-77, why mixed with diatomaceous earth?

4.       In Fig.1, a, b and c, there was no error bar.

5.       Is the relation between monosaccharides and antioxidative ability? Please explain it.

6.       The antioxidative ability and immune promotion activity of OJZ8 and OJC8 were higher. The authors should further explain the results.

7.       In 3.6, what is the result of fru which is the main monosaccharide?

8.       In Conclusion, how to use the Fru to evaluate Maidong? The manuscript did not show the results.

9.       In references, the titles of the journals should be unified.

Author Response

  1. Thanks for the reviewer's advice. The abstract has been carefully revised. A novel quality evaluation strategy based on polysaccharides from MaiDong was emphasized.
  2. Thanks for the reviewer's suggestion, and some current methods for evaluating quality of MaiDong have been added, lines 50-56.
  3. Thanks for the reviewer’s comments. Because MaiDong is sticky herbal material, it is difficult to grind directly. Inert diatomaceous earth was used to avoid clogging during grounding, lines 81-82.
  4. Thanks for the reviewer's comments. The data were average of two measurements, so there were no error bars.
  5. Thanks for the reviewer’s comments. Some discussion was added, lines 217-219.
  6. Thanks for the reviewer's comments, and more discussion was added, lines 217-219 and 247-250.
  7. Thanks for the reviewer’s comments. Fru, the richest monosaccharide in MaiDong polysaccharides, had significantly negative relationship to antioxidant and immunological activity of the polysaccharides, lines 292-293.
  8. Thanks for the reviewer’s comments. Some words were added, lines 339-342.
  9. Thanks for the reviewer’s comments. Done.

Reviewer 2 Report

This is an important research, especially because of the amount of antioxidants and immune activity provided by the Chinese medicinal tonic. Microwave extraction of polysaccharides helps to maintain the tonic's qualities.
The introduction can be improved and put the importance of the use of microwaves in the extraction of polysaccharides.
The analysis methods for the characterization of the samples determined the molecular weight, molar concentration, antioxidant and immune activity, which allows to know the pharmacological activity of the tonic.
The conclusions can be improved in relation to the objectives and goals met in the research project.
This methodology can help to select the best culture, the best variety, as well as the use and quality control of MaiDong root.

Author Response

Thanks for the reviewer’s support and comments. Revision has been made according to the reviewer’s suggestion, lines 59-63, 339-342.

All items mentioned in the manuscript have been revised according to the reviewer’s suggestion.

Round 2

Reviewer 1 Report

This manuscript is seriously corrected. In my opinion, it can be accepted.